# Enhancement of Radiosurgical Treatment Outcome Prediction Using MRI Radiomics in Patients with Non-Small Cell Lung Cancer Brain Metastases

**DOI:** 10.3390/cancers13164030

**Published:** 2021-08-10

**Authors:** Chien-Yi Liao, Cheng-Chia Lee, Huai-Che Yang, Ching-Jen Chen, Wen-Yuh Chung, Hsiu-Mei Wu, Wan-Yuo Guo, Ren-Shyan Liu, Chia-Feng Lu

**Affiliations:** 1Department of Biomedical Imaging and Radiological Sciences, National Yang Ming Chiao Tung University, Taipei 11221, Taiwan; wl0315178315875@ym.edu.tw; 2Department of Neurosurgery, Neurological Institute, Taipei Veterans General Hospital, Taipei 11217, Taiwan; cclee12@vghtpe.gov.tw (C.-C.L.); hcyang3@vghtpe.gov.tw (H.-C.Y.); 3School of Medicine, National Yang Ming Chiao Tung University, Taipei 11221, Taiwan; hmwu@vghtpe.gov.tw (H.-M.W.); wyguo@vghtpe.gov.tw (W.-Y.G.); 4Brain Research Center, National Yang Ming Chiao Tung University, Taipei 11221, Taiwan; 5Department of Neurological Surgery, University of Virginia Health System, Charlottesville, VA 22908, USA; cc5hx@virginia.edu; 6Department of Neurosurgery, Kaohsiung Veterans General Hospital, Kaohsiung 81362, Taiwan; wychung1955@gmail.com; 7Department of Radiology, Taipei Veterans General Hospital, Taipei 11217, Taiwan; 8Department of Nuclear Medicine, Cheng Hsin General Hospital, Taipei 11220, Taiwan; rsliu@vghtpe.gov.tw; 9Molecular and Genetic Imaging Core, Taiwan Animal Consortium, Taipei 11571, Taiwan; 10Institute of Biophotonics, National Yang Ming Chiao Tung University, Taipei 11221, Taiwan

**Keywords:** non-small cell lung cancer, brain metastasis, Gamma Knife radiosurgery, outcome prediction, radiomics, machine learning, magnetic resonance imaging

## Abstract

**Simple Summary:**

Non-small cell lung cancer (NSCLC) is the most common cause of brain metastasis (BM). Approximately 50% of patients with metastatic NSCLC harbor BMs. Within the past decade, Gamma Knife radiosurgery (GKRS) has become one of the first-line treatments for BMs. Ability to predict treatment response after GKRS can therefore guide treatment strategy. This study aimed to determine whether pre-radiosurgical neuroimaging radiomics can predict survival and local tumor control after GKRS. Based on the collected magnetic resonance images and clinical characteristics of the 237 NSCLC BM patients with BMs (for survival prediction) and 256 NSCLC patients with 976 BMs (for prediction of local tumor control), we concluded that the identified radiomic features could provide valuable additional information to enhance the prediction of BM responses after GKRS. The proposed approach provided physicians with an intuitive way to predict the patient outcome based on pre-radiosurgical magnetic resonance images.

**Abstract:**

The diagnosis of brain metastasis (BM) is commonly observed in non-small cell lung cancer (NSCLC) with poor outcomes. Accordingly, developing an approach to early predict BM response to Gamma Knife radiosurgery (GKRS) may benefit the patient treatment and monitoring. A total of 237 NSCLC patients with BMs (for survival prediction) and 256 patients with 976 BMs (for prediction of local tumor control) treated with GKRS were retrospectively analyzed. All the survival data were recorded without censoring, and the status of local tumor control was determined by comparing the last MRI follow-up in patients’ lives with the pre-GKRS MRI. Overall 1763 radiomic features were extracted from pre-radiosurgical magnetic resonance images. Three prediction models were constructed, using (1) clinical data, (2) radiomic features, and (3) clinical and radiomic features. Support vector machines with a 30% hold-out validation approach were constructed. For treatment outcome predictions, the models derived from both the clinical and radiomics data achieved the best results. For local tumor control, the combined model achieved an area under the curve (AUC) of 0.95, an accuracy of 90%, a sensitivity of 91%, and a specificity of 89%. For patient survival, the combined model achieved an AUC of 0.81, an accuracy of 77%, a sensitivity of 78%, and a specificity of 80%. The pre-radiosurgical radiomics data enhanced the performance of local tumor control and survival prediction models in NSCLC patients with BMs treated with GRKS. An outcome prediction model based on radiomics combined with clinical features may guide therapy in these patients.

## 1. Introduction

Non-small cell lung cancer (NSCLC) is a common malignant cancer with a high mortality rate. Approximately 50% of patients with metastatic NSCLC harbor brain metastases (BMs) [1]. Whole-brain radiotherapy, stereotactic radiosurgery, and surgical resection are common treatments for BMs [2,3,4]. Within the past decade, Gamma Knife radiosurgery (GKRS) has become one of the first-line treatments for BMs [5,6]. GKRS has demonstrated efficacy in randomized controlled trials for a limited number of BMs, achieving local tumor control rates of 70 to 80% [7]. However, variations in treatment response exist. Therefore, accurate prediction of such a response may help guide management. Several molecular biomarkers have been associated with local tumor control after GKRS for NSCLC BMs. The mutation status of Epidermal Growth Factor Receptor (EGFR) was suggested to be associated with the local tumor control of BM after GKRS. Lee et al. reported that the local control rate in the EGFR mutant group was approximately 3-fold higher than that in the wild-type group during the 2-year follow-up period after GKRS [8]. Moreover, the insulin-like growth factor I receptor pathway was reported to be over-expressed in BMs and induced treatment resistance in non-small cell lung cancer cells [9,10]. Despite this, the use of molecular biomarkers for predictions necessitates tissue sampling and molecular sequencing, which present additional costs and risks. Therefore, there is a need for accurate prediction tools that utilize readily available clinical and neuroimaging data.

Magnetic resonance imaging (MRI) is a standard-of-care neuroimaging modality in the workup and diagnosis of BMs in NSCLC patients. Some of the imaging features, including size, number, and presence of hemorrhage, have been associated with local tumor control after GKRS [11,12,13]. Nevertheless, the predictive performances of models using conventional imaging characteristics have been limited by these rudimentary tumor descriptions. Radiomics, a method that extracts high-throughput quantitative features from radiographic medical images, has been utilized in cancer research to improve diagnostic, prognostic, and predictive accuracies [14,15,16]. More recently, several studies have suggested that MRI radiomics could facilitate several clinical applications, such as lesion classification, cancer staging, and survival prediction in various types of cancers [17,18,19]. Concurrently, machine learning methods have been widely applied to process high-dimensional data and develop prediction models in radiomics studies [20]. Prior studies have attempted to combine MRI radiomics and machine learning methods to predict response after GKRS in patients with BMs [21,22,23]. However, these studies were limited by small sample sizes and unstandardized radiomics processing.

In this study, we developed local tumor control and overall survival (OS) prediction models using machine learning. Clinical and radiomics data were used in the derivation of these models, and different models were compared. We hypothesized that imaging features extracted from pre-radiosurgical MRIs can improve local tumor control and OS prediction models for NSCLC patients with BMs treated with GKRS.

## 2. Materials and Methods

### 2.1. Patient Cohort

A database of 307 NSCLC patients with BMs treated with GKRS at Taipei Veterans General Hospital between 2011 and 2018 was retrospectively collected. Inclusion criteria for the study: (1) a diagnosis of NSCLC confirmed by lung biopsy or open surgery; (2) presence of ≥1 BM(s) on MRI; (3) underwent GKRS treatment for the BM(s); (4) at least one clinical and neuroimaging follow-up. The flowchart of patient recruitment is shown in Figure 1. The study was approved by the Institutional Review Board, and informed consent was waived.

### 2.2. MRI Preprocessing and Radiomics Feature Extraction

Pre-GKRS MRIs were acquired from each patient, including T1-weighted (T1w; TR/TE = 500/9 ms), contrast-enhanced T1-weighted (T1c; TR/TE = 500/9 ms) and T2-weighted (T2w; TR/TE = 4000/109 ms) images. All the pre-treatment MRI data were acquired on the same day of GKRS followed by the treatment planning and treatment delivery. Pre-processing was applied to improve the reliability of radiomics analysis. Image resolution adjustments were performed to re-sample all voxel sizes to 1 × 1 × 1 mm^3^ for each MRI sequence. The T2w and T1w images were then co-registered to the T1c images using a rigid body transformation followed by a six-parameter rigid body transformation and mutual information algorithm. Afterward, the co-registration quality was visually verified on the processing platform. Finally, image intensities were transformed into standardized ranges (Z-score transformation) based on the whole-image mean and standard deviation for each image set.

The BMs were delineated on pre-radiosurgical MRIs based on a consensus of a multidisciplinary team comprising experienced neurosurgeons and neuroradiologists for GKRS treatment planning. Radiomic features, including histogram, geometry (shape and size), and texture analyses, were extracted from the pre-radiosurgical MRIs. The histogram features describe the global distribution of the region of interest (ROI), such as energy, entropy, maximum, kurtosis, and skewness. The geometry features describe the tumor volume, surface area and shape, and their ratios. The texture features describe the heterogenetic of ROIs based on the gray level co-occurrence matrix (GLCM), gray level run length matrix (GLRLM), and local binary pattern (LBP) [24,25,26,27]. In the feature extraction process, the feature aggregation of GLCM and GLRLM values was performed by averaging over 3D directional matrices to improve rotational invariance [28]. The LBP features were calculated slice-by-slice followed by the histogram analysis of LBP matrices across all slices. The wavelet features were calculated using a three-dimensional wavelet transform function (coif1 wavelet). Wavelet decomposition was conducted by applying low (L) and high (H) pass dimensional filters along three image axes, resulting in eight decomposed image sets: LLL, LLH, LHL, LHH, HLL, HLH, HHL and HHH filtered images. Wavelet decomposition was performed according to the IBSI guidelines for three-dimensional separable wavelets applied to radiomics [29]. For the original MRI and each wavelet image set, 16 histograms and 49 texture features (including GLCM, GLRLM, and LBP) were extracted, yielding 585 features. Eight geometry features were calculated to further quantify the three-dimensional geometry of the ROIs. A total of 1763 MR radiomic features (585 features × 3 image contrasts + 8 shape and size features) were generated for each BM. All the image preprocessing steps and subsequent radiomics extraction were performed using previously published MR radiomics platform (MRP) [30,31] complied with the Image Biomarker Standardization Initiative (IBSI) [28]. A total of 1763 radiomic features were extracted from the pre-radiosurgical MRIs. The diagram of the radiomics workflow is displayed in Appendix A. The formulae for the calculation of radiomics are listed in Appendix A.

### 2.3. Feature Selection and Classification Models

In this study, classification models were developed to predict local tumor control and OS after GKRS. The BM response to GKRS was assessed by comparing the last MRI follow-up in patients’ lives to the pre-GKRS MRI and categorized as follows: (1) poor tumor control (tumor progression): more than 10% increase in tumor volume and (2) good tumor control (stable tumor volume or regression): less than 10% increase or more than 10% decrease of tumor volume. For the OS prediction, we applied the median survival to categorize patients into good or poor survival groups. A hold-out method was used to divide the dataset into a training dataset (70% of the total samples) and a testing dataset (the remaining 30%). Feature selection and construction of predictive models were applied to the training set. The testing set was used to assess the performance of the derived models.

Three predictive models for each outcome were developed using the following features: (1) clinical features (i.e., Karnofsky performance status (KPS), presence of extracranial metastases, treatment of primary NSCLC, number of BMs, and volume of BMs); (2) radiomic features; and (3) the combination of clinical and radiomic features. A two-step feature selection was adopted to identify key radiomic features and reduce feature redundancy. The first step was conducted using a two-sample *t*-test to identify the feature candidates with significant differences between groups (good vs. poor for local tumor control; and good vs. poor for survival). For each classification model, 25 features with the smallest *p*-value (<0.05) were first identified separately. Final key features for model training were further sieved out by using the sequential forward selection (SFS) algorithm for each outcome prediction [32].

The sample sizes of the good and poor local tumor control groups were unbalanced. To reduce the bias, a near miss under-sampling method (NearMiss-2) was used to equalize the number of samples in two classes [33]. For the prediction of local tumor control, the proposed model was constructed by the lesion-wise approach. For the prediction of overall survival, the model was developed using the patient-wise analysis, and only the radiomic features extracted from the largest BM were used in the model derivation. The rationales that we only focused on the largest BM included: (1) to produce a more reliable estimate of radiomic features with a sufficient number of voxels [34]; (2) to mostly represent the malignancy of BM and impact on patients’ survival; (3) to alleviate the potential variability of radiomics features extracted from multiple lesions within a patient. Features identified through the two-step feature selection process were used in the model training, and support vector machines (SVM) with Gaussian function kernel and Bayesian optimization for hyperparameters were trained for each prediction.

### 2.4. Model Performance and Statistics

The performance of three models for each outcome prediction based on different features were evaluated by calculating the accuracy, sensitivity, specificity, and area under the receiver operating characteristic curve (AUC) in the testing dataset. To further statistically compare the performance between three models for each outcome prediction, we employed bootstrap random resampling 100 times on the testing dataset and repeatedly evaluate model performance for each sampling dataset [35]. Based on the evaluated variation for each model performance, we used the paired *t*-test to identify the significant differences of accuracy, sensitivity, specificity, and AUC between the predictive models with Bonferroni correction for the multiple comparisons.

## 3. Results

### 3.1. Clinical Characteristics of Patients

The clinical characteristics of finally recruited 237 patients with BMs (for the survival prediction) and 256 patients with 976 BMs (for the prediction of local tumor control) are listed in Table 1. All the 237 patients have complete information of OS (recorded time-to-death event) and clinical data, and all the 976 BMs have complete information of local tumor control and clinical data.

### 3.2. Selected Radiomic Features

For the prediction of local tumor control after GKRS, 25 radiomic features were first selected (Appendix A). Five final radiomic features were selected by the SFS algorithm from the 25 features identified. These features included the textural features focusing on the informational coefficient of correlation in T1w and T1c images, and the local homogeneity in T1w images. The final selected features of local tumor control prediction are listed in Table 2.

For OS prediction after GKRS, a total of 25 features were first selected (Appendix A). Four final radiomic features were selected by the SFS algorithm from the 25 features identified. These features included the histogram and textural features focusing on the minimum and maximum of intensity distribution, the local density of voxels, and the correlation of neighboring intensities in T1w images. The final selected features of OS prediction are listed in Table 2.

### 3.3. Performance of Outcome Prediction Models

Figure 2 demonstrates the workflow of prediction model development and validation in this study. For the local tumor control prediction, the combined model (derived from clinical and radiomics data) achieved the best AUC of 0.95, with a sensitivity of 91%, a specificity of 89%, and an accuracy of 90% in the testing dataset. Table 3 lists the detailed performance for each model and statistical comparisons between different models. The receiver operating characteristic (ROC) curves of the three models are illustrated for the prediction of local tumor control in Figure 3a and OS in Figure 3b, respectively. 

The predictive model scores of local tumor control and OS-derived clinical and radiomics data in the testing set are illustrated in Figure 3c,d, respectively. Representative cases illustrating the relationship between radiomic values and local tumor control are shown in Figure 4a, and the relationship between radiomic values and OS in Figure 4b.

## 4. Discussion

Prediction of local tumor control and OS for NSCLC patients with BMs after GKRS is challenging in clinical practice. In this study, we proposed a radiomic-based approach to predict treatment outcomes after GKRS in these patients. Radiomic features extracted from pre-radiosurgical MRIs enhanced the prediction models derived from clinical characteristics alone.

Previous radiomic studies predicting local control of BM after GKRS were limited by their relatively small sample sizes and non-standardized radiomics procedure. For example, Mouraviev et al. included 87 patients with 408 BMs and trained a random forest model based on MRI radiomics, and clinical features to achieve an AUC of 0.79 in predicting local control after GKRS [21]. In this study, we developed a prediction model of local tumor control using both clinical and radiomic data, which achieved an AUC of 0.95. The superior performance of our proposed model may be attributed to the following factors. First, the larger sample size in our study may have allowed for better representation of the general cohort and consideration of the potential variations in sampling. Second, in addition to contrast sequences, T1w sequence without contrast enhancement was also acquired, which provided additional information in characterizing tumor compositions. Third, a two-step feature selection was performed in this study, comprising two-sample *t*-tests to remove a large number of non-significant redundant features followed by the application of the SFS algorithm to identify the final key features. An appropriate feature selection can reduce redundancy and improve the accuracy of the classifier with high computing efficiency. Finally, the image pre-processing and feature extraction methods applied in this study were in accordance with the IBSI recommendations. IBSI provided a comprehensive review and suggestions of each essential step in radiomics analysis, including the image pre-processing, lesion segmentation, feature extraction, and validation [28,36]. The standardization of radiomics in this study could improve the reliability and performance of the prediction models.

For local tumor control prediction, we noted that all five selected features described the extent of tumor heterogeneity which were associated with tumor malignancy and aggressiveness [37]. Four of the five selected features were associated with the informational measure of correlation 1 (IMC1) feature. The results showed that BMs with poor local tumor control exhibited higher values of IMC1. This phenomenon implied that BMs with high intra-tumor heterogeneity may result in poorer treatment effects in GKRS. Previous studies have reported that the values of texture features may be associated with local failure in BM patients after GKRS [21]. In addition, we observed log-linear correlations (r > 0.9, *p* < 0.001, Appendix A) between IMC1 features and initial tumor volume. Combining our model performance and this logarithm relation, we suggested that IMC1 features may have the potential to predict local tumor control of BMs with similar or subtle changes in tumor volumes. We demonstrated two representative cases with similar volumes and their local tumor control after GKRS (Figure 4a–c). The good local tumor control one presented lower IMC1 and homogeneity 1 values than the case with poor local tumor control.

For the prediction of local tumor control, we found that all the 9 misclassified lesions of the testing dataset were small BMs ranged from 12 to 141 mm^3^ in volumes (with 5 to 11 mm in maximal 3D diameters). In another hand, if we only focused on the small BMs with volumes less than 150 mm^3^ in the test dataset (N = 58), the proposed radiomic and clinical feature-based classification models could achieve an accuracy of 76% and 64%, respectively. The combined model based on both radiomic and clinical features for the small BM prediction could still improve the accuracy to 85% (the accuracy of the full test dataset composed of both small and larger BMs was 90%). In summary, despite all the misclassified lesions are small BMs for the prediction of local tumor control, not all of the small BMs were misclassified (in fact, the accuracy could still achieve 85%). These results revealed that even the small BMs may be relatively hard to acquire accurate imaging features due to limited lesion volumes, the added values of radiomic features to enhance the prediction of local tumor control could still be observed in small BMs.

Clinical characteristics included in this study, such as the KPS, extracranial metastases, and the number of lesions were associated with OS in these patients [38]. We combined radiomics with these clinical predictors to derive an OS prediction model. The combination of clinical and radiomic data improved the performance of the model to achieve an AUC of 0.81 compared to a model based on radiomics (AUC of 0.66) or clinical data (AUC of 0.75). Patients with longer OS had higher maximum, cluster tendency, correlation values, and lower minimum values in the T1w image. The intensity of the T1w signal was related to tumor composition. Tumors with high intracellular mucin usually have lower T1w signal intensity due to their high water content [39] and are susceptible to secondary electron ionization inducing DNA damage [40]. High values of T1w correlation and cluster tendency in BMs may suggest high intratumoral homogeneity reported to associate with more appropriate dose control of tumor margin [41]. All selected radiomic features were extracted from pre-contrast T1w images which may reflect that the composition of tumor were the potential predictors for the survival of BM patients. Two representative patients with similar-sized BMs illustrate the relationships between clinical and radiomic features and OS (Figure 4d–f). Case #3 with better KPS (>90) and without extracranial metastases had better survival than Case #4 with poorer KPS (<90) and with extracranial metastases (22.6 vs. 6.8 months). Compared to the local tumor control prediction model (AUC of 0.95), the OS prediction model achieved a lower AUC of 0.81, indicating that more comprehensive information with regard to the clinical and genetic characteristics may be necessary for predicting survival.

The limitations of the study should be recognized. First, although the dataset acquired from a single institution may improve the homogeneity of the data [21], the external validation dataset should be considered in future studies to improve the generalizability of the prediction models. Second, radiomic features extracted from the largest BM in each patient were used in our model derivation, as previous studies suggested a sufficient number of voxels provides reliable results [34,42]. However, the optimal approach in quantifying radiomic features from multiple BMs remains debatable, and additional studies to explore this effect may benefit the prediction of OS in BM. Third, additional treatments, including neurosurgery, whole-brain radiotherapy, tyrosine kinase inhibitor treatment, and chemotherapy, were delivered under the physicians’ prescription based on each patient’s condition reflecting the issue of complicated clinical practice. Further studies are required to investigate the combined effects of these additional treatments. Finally, based on the newest update of the Graded Prognostic Assessment for lung cancer using molecular markers (Lung-molGPA), the EGFR and anaplastic lymphoma kinase mutation status were included as prognostic factors for patients with NSCLC and BMs [43]. In our study, only a proportion of patients had reported EGFR status. Future studies utilizing clinical, MRI radiomic, and molecular information to build more comprehensive models should be encouraged.

## 5. Conclusions

The inclusion of MRI radiomics in local tumor control and OS prediction models improves the accuracies and reliabilities of these models. Informational measures of correlation and homogeneity values of MRIs may suggest tumor radio-resistance. Histogram features, correlation, and cluster tendency extracted from MRIs combined with clinical characteristics can also improve OS prediction. The dynamic changes in radiomic features during follow-up may be worthy of further investigations to guide treatment strategies in these patients.

## Figures and Tables

**Figure 1 cancers-13-04030-f001:**
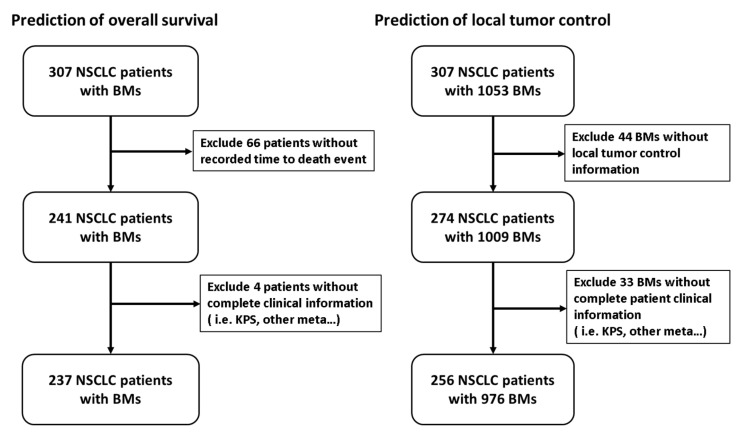
Flowchart of patient recruitment.

**Figure 2 cancers-13-04030-f002:**
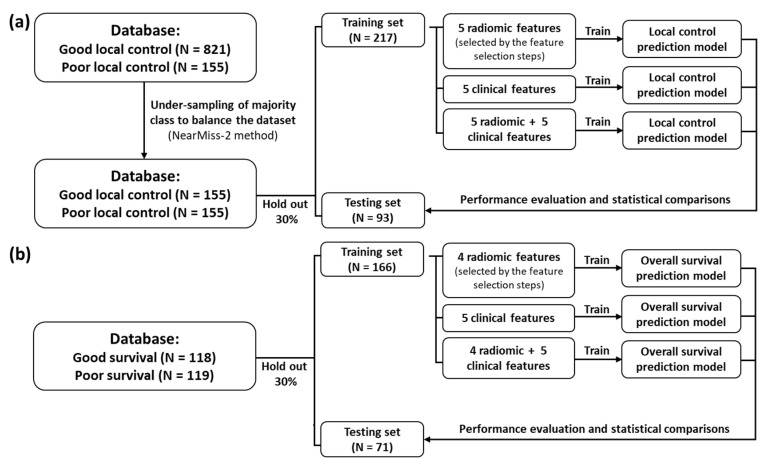
Flowchart for development and validation of machine learning model. The data analysis process for (**a**) local tumor control and (**b**) overall survival prediction.

**Figure 3 cancers-13-04030-f003:**
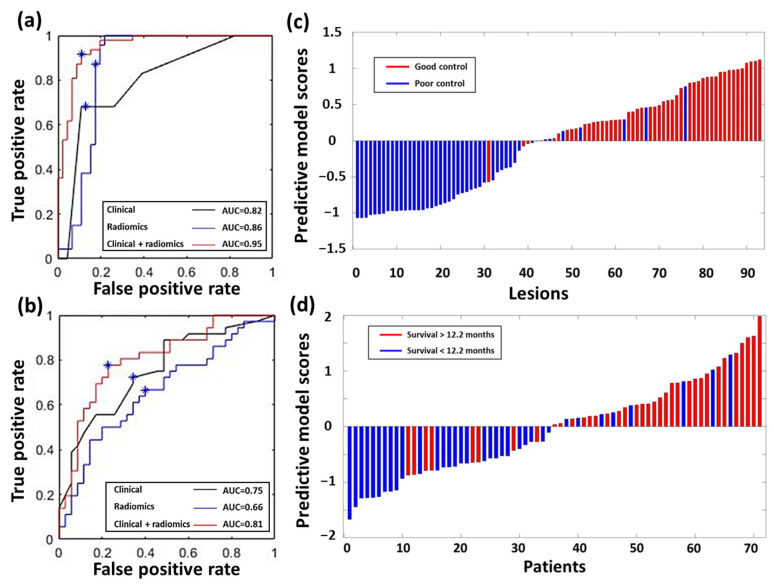
Results of the SVM models for GKRS treatment outcome prediction. Receiver operating characteristic curves of constructed models for predicting (**a**) local tumor control and (**b**) OS estimated based on the testing dataset. The blue asterisks represent the optimal parameter setups. The corresponding predictive model scores of each case for the (**c**) local tumor control and (**d**) OS prediction. The red bars in (**c**,**d**) represent the cases with good outcomes (good local tumor control and longer OS), and the blue bars represent the cases with poor outcomes (poor local tumor control and shorter OS). Most of the red bars exhibit positive predictive model scores, and most of the blue bars show negative values, indicating satisfactory prediction performance.

**Figure 4 cancers-13-04030-f004:**
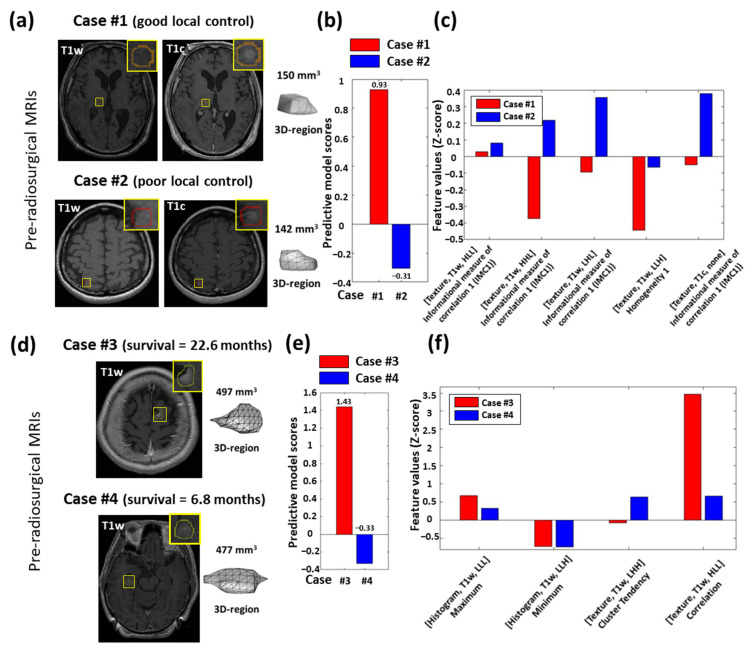
Representative cases for the predictions of local tumor control (Case #1 and #2) and overall survival (Case #3 and #4). (**a**) Case #1 with a measured BM volume of 150 mm^3^ (upper) shows good local tumor control after GKRS, and Case #2 with a measured BM volume of 142 mm^3^ (lower) shows poor local tumor control after GKRS. (**b**) The predictive model scores are estimated by the proposed prediction model based on both radiomic and clinical features. (**c**) The key selected radiomic features for Cases #1 and #2. (**d**) Case #3 with a measured BM volume of 497 mm^3^ (upper) presents better overall survival (22.6 months) after GKRS, and Case #4 with a measured BM volume of 477 mm^3^ (lower) shows poor survival (6.8 months). (**e**) The predictive model scores are estimated by the proposed prediction model based on both radiomic and clinical features. (**f**) The key selected radiomic features for Cases #3 and #4. The yellow boxes in the upper-right corner of (**a**,**d**) are the zoom-in views of MRIs in the lesion areas.

**Table 1 cancers-13-04030-t001:** Patient and BM characteristics.

Characteristic	Value	Percentage or Range
***Patients for survival prediction (N = 237)***		
Age	60.8	22.6–91.3
Gender (Male:Female)	115:122	
Overall survival (Month)	12.2	0.07–64.7
**Other metastasis**		
Yes	117	49.4%
No	120	50.6%
**KPS**		
≥90	164	69.2%
<90	73	30.8%
**Original tumor control**		
Yes	108	48.8%
No	129	50.4%
**Number of tumors**		
≥3	123	51.9%
<3	114	48.1%
**NSCLC histology**		
Pure adenocarcinoma	233	98.4%
Adenocarcinoma + Large cell carcinoma	1	0.4%
Adenocarcinoma + Squamous cell carcinoma	1	0.4%
Undifferentiated NSCLC	2	0.8%
**Additional treatments**		
Neurosurgery	22	9.3%
Whole-brain radiotherapy	30	12.7%
Tyrosine kinase inhibitor	207	87.3%
Chemotherapy	137	57.8%
***BMs for prediction of local tumor control (N = 976)***		
**Local tumor control**		
Good	821	84.1%
Poor	155	15.9%
**Maximum 3D diameter (d)**		
0 < d < 5 mm	11	1.1%
5 < d < 10 mm	410	42.0%
10 < d < 20 mm	416	42.6%
d > 20 mm	139	14.3%
**Median GK dose (Gy)**		
Tumor center	28.6	18.7–50
Tumor periphery	19	12–30

**Table 2 cancers-13-04030-t002:** Final selected radiomic features for the outcome prediction.

Image Contrast	Wavelet Filtering	Radiomics Type	Feature Name	Outcome Status	
				Good	Poor
***Prediction of local tumor control***					
T1w	LLH	Texture-GLCM	Homogeneity 1	−0.52 ± 0.46	0.53 ± 1.13
T1w	LHL	Texture-GLCM	Informational measure of correlation 1 (IMC1)	−0.51 ± 0.49	0.52 ± 1.12
T1w	HLL	Texture-GLCM	Informational measure of correlation 1 (IMC1)	−0.52 ± 0.46	0.53 ± 1.12
T1w	HHL	Texture-GLCM	Informational measure of correlation 1 (IMC1)	−0.54 ± 0.45	0.55 ± 1.11
T1c	none	Texture-GLCM	Informational measure of correlation 1 (IMC1)	−0.50 ± 0.49	0.52 ± 1.12
***Prediction of overall survival***					
T1w	LLL	Histogram	Maximum	0.32 ± 1.03	−0.33 ± 0.86
T1w	LLH	Histogram	Minimum	−0.30 ± 1.17	0.31 ± 0.67
T1w	LHH	Texture-GLCM	Cluster Tendency	0.19 ± 1.17	−0.20 ± 0.76
T1w	HLL	Texture-GLCM	Correlation	0.01 ± 0.96	−0.01 ± 1.06

GLCM: gray-level co-occurrence matrix. In the column of wavelet filtering, L represents a low-pass filter, and H represents a high-pass filter. The combination of L and H letters stands for the filter type applied to the three image axes in order.

**Table 3 cancers-13-04030-t003:** Statistical comparisons between developed predictive models.

Model Performance	Radiomics	Clinical	Combined	*p*-Values		
				Radiomics vs. Clinical	Radiomics vs. Combined	Clinical vs. Combined
***Prediction of local tumor control***						
AUC	0.86 ± 0.10	0.80 ± 0.08	0.95 ± 0.09	<0.001 *	<0.001 *	<0.001 *
Accuracy	0.85 ± 0.10	0.76 ± 0.09	0.89 ± 0.11	<0.001 *	<0.001 *	<0.001 *
Sensitivity	0.85 ± 0.17	0.69 ± 0.14	0.87 ± 0.17	<0.001 *	0.124	<0.001 *
Specificity	0.85 ± 0.11	0.83 ± 0.10	0.91 ± 0.12	0.118	<0.001 *	<0.001 *
***Prediction of overall survival***						
AUC	0.64 ± 0.24	0.78 ± 0.15	0.82 ± 0.15	<0.001 *	<0.001 *	<0.001 *
Accuracy	0.62 ± 0.21	0.71 ± 0.10	0.80 ± 0.17	<0.001 *	<0.001 *	<0.001 *
Sensitivity	0.68 ± 0.25	0.73 ± 0.25	0.77 ± 0.14	0.046	<0.001 *	0.006 *
Specificity	0.55 ± 0.29	0.68 ± 0.20	0.81 ± 0.24	<0.001 *	<0.001 *	<0.001 *

* Significant difference is identified based on the paired *t*-test with Bonferroni correction.

## Data Availability

All data generated or analyzed during this study are included in this published article and its Appendix A. To protect patient privacy, the raw images and data collected in this study can be only accessed by contacting the corresponding author (C.-F.L.).

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
