# Peer review of "Enhancement of Radiosurgical Treatment Outcome Prediction Using MRI Radiomics in Patients with Non-Small Cell Lung Cancer Brain Metastases"

_cancers, 2021, doi:10.3390/cancers13164030_

Round 1

Reviewer 1 Report

The study by Liao and colleagues from the renowned research group of Prof. Lu demonstrates the ability of MRI radiomics to predict treatment outcome in patients with non-small cell lung cancer brain metastases after gamma knife radiosurgery. The manuscript is well written and the methodology is carefully chosen and described. The results are convincing and support the conclusions. In addition, the authors reflect on the few limitations of the study. I have only one question for the authors: 

In our experience, the size of the metastases plays a major role in the success of radiomics models. Especially for small metastases, the models often work very poorly. Could the authors also find such a correlation in their model? Are the misclassified lesions predominantly small ones? Please comment.

Reviewer 2 Report

Abstract – More clearly define the predicted outcomes. Is this local tumor control and survival at a certain timepoint (ex 6 months, 1 year, etc), a time to event prediction, or a final status prediction?

Introduction – How well do these molecular biomarkers for treatment response in BMs  perform (page 2 line 62)?   

Materials and Methods –

How far before treatment were MRIs obtained? Was this consistent?

Were all patients treated definitively using GKRS or were some courses adjuvant to other therapy (ex surgery)?

Was skull-stripping performed?

Have the authors considered using a time to death prediction approach rather than a good vs poor survival group prediction?

The description of the two experiments (local tumor control and survival) given in lines 155 to 165 is unclear. Was only the largest lesion in cases with multiple lesions used in each task? If so, was there any consideration of performing a lesion-wise or multi-lesion analysis rather than using the tumor with the greatest volume for patient level survival prediction? If lesions were considered individually for local tumor control, please specify. Furthermore, was there any variation in results between patients with multiple lesions and patients with fewer lesions?

At what time point is local tumor control determined?

Results

Is there significance to the scale of the model scores given on the y-axis for Figure 3 (eg does a score of -2 represent a 100% more probabilistic score than a -1 for poor survival)?

It is not entirely clear what is meant to be observed in the zoomed in boxes in the upper right hand corner of MRIs in Figure 4. Is this meant to simply be a zoomed in view or are radiomic features visualized?

Again, in Figure 4 please clarify the significance of the y-axis values for predictive scores. Here it appears that the negative scores predicted in (e) and (b) are actually quite low in magnitude when viewing Figure 3 (c) and (d).

Reviewer 3 Report

In general, this study represents an interesting data on a novel radiological approach to predict overall survival and local progression in radiosurgically treated NSCLC BM patients. In comparison to previous data, the statistical comparisons between the developed predictive models display high significant AUC, Accurarcy and Specificity for local tumor control and OS in a larger cohort of BM patients. Indeed, I have some questions/suggestions:

1. Are there any differences in outcomes between the primary tumor histology, f.e. adenocarcinoma vs. squamous cell carcinoma? 

2. Did you investigate if oncological therapies, such as immunotherapy or targeted therapy, influenced the OS or local progression? Previous literature have shown that the combination of GKRS and immunotherapy and/or targeted therapy are associated with a longer survival in NSCLC BM patients. Were there any differences of radiomic features between patients with and without IT?

3. How was "poor outcome" exactly define? 
